# OpenReview forum: "What Information Matters? Graph Out-of-Distribution Detection via Tri-Component Information Decomposition"
_ICML.cc/2026/Conference — ICML 2026 regular_

### Official Review · Reviewer_jWQo · 2026-03-05

**Soundness:** 2
**Presentation:** 3
**Significance:** 2
**Originality:** 3
**Overall Recommendation:** 4
**Confidence:** 4

**Summary:**

In this paper, the authors propose a novel method, namely TIDE, to address out-of-distribution (OOD) shifts (feature shift, structural shift, or joint shift) in node classification tasks. Its key idea is to decompose the predictive information of graph data into feature-specific, structure-specific, and joint-input components. It preserves label-relevant joint information through the Information Bottleneck (IB) objective while filtering out spurious correlations and irrelevant noise, and strengthens information separation via conditional independence regularization and pairwise mutual information minimization.

**Compliance With Llm Reviewing Policy:**

Affirmed.

**Final Justification:**

The authors have solved all of my issues. So I choose to accept this paper.

**Key Questions For Authors:**

Please refer to the Weaknesses. If the authors can provide reasonable explanations in the rebuttal, I will consider raising my score.

**Limitations:**

No. The authors have not adequately discussed limitations and potential negative societal impacts.

**Strengths And Weaknesses:**

Strengths

1. The organization of this paper is clear.

2. The authors introduce a clear tri-component decomposition perspective for graphs.

3. In experiments, the authors evaluate the baseline models on seven datasets.

Weaknesses

1. The transition from the exact decomposition $I(X, A; Y) = I(Z; Y) + I(X; Y | Z) + I(A; Y | Z)$ to the implemented surrogate $I(X, A; Y) ≈ I(Z; Y) + I(V; Y) + I(Q; Y)$ is not theoretically justified. $I(V; Y)$ and $I(Q; Y)$ are not, in general, equal to $I(X; Y | Z)$ and $I(A; Y | Z)$. The paper does not provide conditions under which this substitution is valid, beyond regularizing overlaps.

2. In Proposition 4.1’s proof, the authors write $P(z | x, a, y; θ)$, which suggests z depends on y. This contradicts the model description ($Z = f(X, A)$).

3. It is hard for me to accept a Deep Learning method without a released code link. Also, the authors should provide an analysis of the complexity comparisons.

4. The OOD shifts in experiments are relatively mild, and the model's robustness in scenarios with extremely unfamiliar features or topology is not verified.

5. Mutual information estimation is notoriously difficult. The reliance on CLUB (an upper bound) and reconstruction-based surrogates raises stability concerns. The discussion of estimator variance, training stability, or sensitivity to estimator choices is limited.

---

> ### Author Rebuttal · Authors · 2026-03-26
>
> We greatly appreciate the reviewer for your careful reading and insightful comments. We carefully address them as follows:
>
> >W1 On the transition from the exact decomposition to the implemented surrogate.
>
> We agree that, in general, $I(V;Y)\neq I(X;Y|Z)$ and $I(Q;Y)\neq I(A;Y|Z)$. However, Eq.9) is **not** intended as a direct equality substitution, but as a **structured variational decomposition** induced by TIDE’s design (We will revise the misleading notation of = to $\approx$).
>
> Starting from the exact decomposition in Eq.8), the term $I(X;Y|Z)$ is precisely the label information in $X$ **not already captured by $Z$**. TIDE makes $I(V;Y)$ approximate this residual through three mechanisms:
> 1. **Eq.12), IB on $V$**: $\max I(V;Y)-\beta_V I(X;V)$ encourage $V$ to retain as much **label-relevant** information from $X$ as possible under compression;
> 2. **Eq.10), CInd**: enforcing $I(X;A|Z)\to0$ pushes the **shared/joint predictive information** of $X$ and $A$ into $Z$, so what remains outside $Z$ is predominantly single-input residual information;
> 3. **Eq.11), PMI**: minimising $I(Z;V), I(V;Q)$ prevents $V$ from re-encoding information already captured by $Z$ or by each other.
>
> At optimum, with (1-3), the remaining predictive information available to $V$ is precisely the **residual not explained by $Z$**, yielding: $I(V;Y)\approx I(X;Y|Z)$ and analogously $I(Q;Y)\approx I(A;Y|Z)$. We will clarify this in the revision.
> >W2 Prop. 4.1 proof clarification.
>
> We greatly value this feedback in improving the clarity of our work. We agree that the notation used in the current proof is **imprecise and may mislead** the interpretation of $Z$ and $Y$. However, the **proposition itself remains correct**. We will revise the proof based on the following precise and clearer statement.
>
> Stated in Prop. 4.1, $Z$ is encoded from $(X,A)$ by the network $f$. Hence, $Z$ depends only on $(X,A)$ (i.e.,$Z\perp Y|(X,A)$), and the Markov chain $ Y \rightarrow (X,A) \rightarrow Z $ holds. This implies $p(z|x,a,y)=p(z|x,a)$ and we have $I(Z;Y|X,A)=0$. More explicitly, $I(Z;Y|X,A)=\mathbb{E}_{p(x,a)}[D\_{KL}(p(z,y|x,a)||p(z|x,a)p(y|x,a))].$ Under the above Markov chain, $p(z,y|x,a)=p(z|x,a)p(y|x,a)$, so the KL term is zero, which gives $I(Z;Y|X,A)=0.$ Then, by the chain rule of mutual information, $I(X,A,Z;Y)=I(X,A;Y)+I(Z;Y|X,A)=I(X,A;Y).$
>
> Thus, while we acknowledge that the current proof is imprecise, the **proposition and its conclusions remain valid**.
> >W3 On code release and complexity.
>
> We fully support **reproducible research**, reflected by our step-by-step implementation notes in Sec 4/4.3, Appx.G,H, and Alg. 1/2.  Following reviewer's suggestion, we provide the code in this anonymised repo: https://anonymous.4open.science/r/TIDE_ICML_7B1LA/README.md. Regarding complexity, TIDE adds 2 auxiliary networks and MI-based regularisers during training only. Inference uses only the joint encoder $Z$, so test-time cost is essentially the same order as a GCN baseline. The extra cost is thus a training-time constant-factor (under suitable sampling batchsize for MI estimation) overhead. Table 31's **computation-cost** (Appx. P) shows that TIDE incurs $\approx$ 2-4x training-time increase, a small memory increase, and **comparable inference time**, while having substantially better detection performance.
> >W4 On whether the OOD shifts are too mild.
>
> We respectfully note that the evaluation is **broader than a mild shift**. Our setting follows **widely used protocols** from GNNSAFE, NODESAFE, and GOLD, and the chosen shifts are designed to probe different aspects of node OOD. Structure and feature shift test controlled single-input shifts, label-leave-out introduces a harder semantic shift, and Arxiv and Twitch provide realistic joint shifts through temporal and cross-region multi-graph settings. Hence, the benchmark covers both controlled analysis and realistic distribution shifts across multiple difficulty levels.
>
> Due to space limit, we invite the reviewer to review our W1 response to Reviewer hZWj for more extreme scenario discussion.
>
> >W5 On MI estimation stability.
>
> As clarified in the paper, the MI terms are used as **regularisers**, not hard constraints on the decision boundary. The dominant signal for the classifier still comes from the label-likelihood term and the IB-style compression term. The MI regularisers only shape the representation geometry. The regularisers influence are controlled by $\beta$'s (IB terms), $\alpha$’s (PMI), and $\lambda$ (CInd).
>
> **Empirical evidence:**
> * **Appendix L, Tables 6-8:** hyperparameter sweeps over several values show smooth metric changes and a consistent advantage over baseline, suggesting stability of MI estimations.
> * **Appendix L, Table 9:** replacing CLUB with InfoNCE yields similar performance, indicating that TIDE depends on the overall IB-decomposition design rather than any single MI estimator and optimisation remains stable under reasonable perturbations of these approximations.

---

> > ### Author Rebuttal · Reviewer_jWQo · 2026-04-01
> >
> > The authors solved all of my issues about this manuscript. I have raised the score.

---

> > > ### Author Response · Authors · 2026-04-02
> > >
> > > Dear Reviewer jWQo,
> > >
> > > We sincerely thank you for the time you dedicated to providing constructive feedback. We greatly appreciate your recognition that the concerns have been fully addressed.
> > >
> > > If appropriate, we would be very grateful if you would consider supporting our paper during internal discussions or consider a score raise.
> > >
> > > Once again, thank you for your attention to detail, which greatly improves the quality of our manuscript. We will ensure that all clarifications are included in a future revision.
> > >
> > > Authors 1142

---

### Official Review · Reviewer_1QJA · 2026-03-12

**Soundness:** 3
**Presentation:** 3
**Significance:** 3
**Originality:** 3
**Overall Recommendation:** 5
**Confidence:** 3

**Summary:**

This paper introduces TIDE, a novel and effective Tri-Component Information Decomposition framework that explicitly decomposes information into feature-specific, structure-specific and
joint components. By only preserving the label-relevant part of the joint information while filtering out spurious feature- and structure-specific information, their framework can separate the in-distribution and out-of-distribution models with superior performance compared to the baselines,  without hurting the in-distribution utility.

**Compliance With Llm Reviewing Policy:**

Affirmed.

**Final Justification:**

The author resolved almost all my concerns, so i keep my original score.

**Key Questions For Authors:**

See weaknesses.

**Limitations:**

Yes, but the limitations are discussed in the supplementary, i think it would be better to move it to the main body of the paper.

**Strengths And Weaknesses:**

**Strengths:**

1. The paper is well written and clearly organised.
2. The paper provides solid theoretical grounding and clear interpretability analysis, especially the explanation of why IB benefits graph OOD detection(Section 5).
3. The paper presents clear problem formulation and offers a novel perspective for addressing the OOD detection problem in graph domain.

**Weaknesses:**

1. One strong assumption of the method is that feature-specific and structure-specific signals are often spurious correlations. Is this assumption universally valid? There may be cases where these signals contain useful predictive information, and TIDE might discard them. It would be helpful to analyse the predictive power of feature-specific and structure-specific components to better justify this design choice.

2. It would be helpful to provide a clear definition of the difficulty levels mentioned in the caption of Table 1.

3. I notice that in the overall performance of TIDE, the FPR95 values for the Arxiv and Twitch datasets are relatively high. Could the authors elaborate on why this happens? Since Twitch is the only dataset that detects OOD graphs rather than OOD nodes, could it be that TIDE works better for node-level OOD detection than graph-level OOD detection? In addition, the ID accuracy is slightly lower than the baseline, which may also require further analysis.

4. I am confused by the following sentence in Section 6.1: “While datasets like Amazon and Coauthor exhibit strong performance with existing SOTA baselines, driven by high classification accuracy, which yield more discriminative energy scores for OOD detection — TIDE further improves detection performance.” Could the authors clarify this statement? In particular, it would be helpful to explain how the results in Table 1 support this claim. Also, there may be a small mistake here: only the Amazon dataset appears to exhibit stronger baseline performance in Table 1.

---

> ### Author Rebuttal · Authors · 2026-03-26
>
> We thank the reviewer for their recognition of our contributions. We address your comments as follows.
>
> > W1 Clarification on TIDE design
>
> We thank the reviewer for this insightful question. We invite the reviewer to read our W1 response to Reviewer hZWj for a comprehensive explanation, which we highlight as follow.
>
> Our method **does not assume** that feature- and structure-specific signals are universally spurious. Rather, our research focus is that for OOD detection, these single-input signals are often the ones most **vulnerable to distribution shift**, even when they remain predictive in-distribution. This is stated directly in the paper: "although all three signals can be predictive in-distribution," feature- and structure-only correlations can become misleading under shift, whereas the joint feature-structure signal is more shift-indicative for separating ID from OOD. Our goal is to reduce this for improved OOD detection.
>
> This is also **reflected in the empirical results**. Table 1 shows that TIDE maintains competitive ID performance while significantly improving OOD detection across datasets. For example, on Amazon and Coauthor, ID accuracy remains essentially unchanged relative to strong baselines, while OOD metrics still improve; on more challenging settings such as Twitch and Arxiv, TIDE continues to improve detection while keeping ID performance competitive. In addition, Figures 1 and 2 illustrate the core motivation: standard SL mixes joint and single-input signals, while TIDE explicitly isolates feature- and structure-only components so that the main encoder can focus on the more robust joint signal for detection.
>
> > W2 On dataset clarification
>
> As stated in Sec. 6 and Appendix M, the 7 benchmarks include OOD nodes generated via structure manipulation, feature interpolation, label exclusion, or temporal splits, and Twitch uses graphs from different regions as OOD. Thus, the underlying subsets cover structure, feature, label, and temporal/cross-region shifts, which naturally vary in difficulty. We will revise the caption to make this definition explicit.
>
> > W3: Clarification of performance on Arxiv and Twitch
>
> We thank the reviewer for the thoughtful question. First, we clarify a possible misunderstanding: **Twitch is still evaluated as a node-level OOD detection following prior setting**. The difference is that Twitch is a multi-graph real-world shift setting, where ID nodes come from one region graph and OOD nodes come from graphs of other regions, such that difficulty naturally varies depending on  each OOD region graph's similarity to ID. In this setting, the metrics should be interpreted together: FPR95 is a stricter and more local metric: false positive rate at 95% true positive rate. AUROC and AUPR reflect the overall ranking quality across thresholds, and therefore provide a broader view of ID-OOD separability - which Table 1 demonstrates TIDE significant improvements on Twitch.
>
> For Arxiv, the relatively high FPR95 is closely related to the fact that it is a **40-class classification task** with comparatively low ID accuracy. The energy score is defined as $e = -\log \sum_{c} \exp(f_c(x))$, where $f_c(x)$ is the logit for class $c$. This means the energy score is formed by aggregating the logits across all classes. As the number of classes increases, or when some classes cannot be accurately predicted, the logits become less separable. The resulting ID/OOD energy values thus becomes less distinguishable, which makes detection harder. This is consistent with our analysis in Sections 5 and 6.3, and in prior work (e.g., GOLD), where we explain that sharper predictive distributions lead to more separated energy scores, while lower classification quality reduces the discriminativeness of energy-based detection.
>
> > W4 On the connection between high ID accuracy with improved OOD detection
>
> Thank you for pointing this out. Our intended meaning in Sec. 6.1 is that higher classification accuracy (Amazon and Coauthor Acc > 93%) typically produces **lower predictive entropy and sharper logits, which in turn makes the energy score more discriminative**. Following the previous explanation, since energy is defined as an aggregation of logits, better-separated logits induce a clearer distinction between confident ID and uncertain OOD samples. This is precisely the theory-practical connection made in Sec. 5 and Sec. 6.3, where the paper explains that lower entropy on ID and higher entropy on OOD translate into lower energies for ID nodes and higher energies for OOD nodes.
>
> Additionally, the sentence is not a mistake. While Amazon is clearer, the baseline performance for Coauthor is also strong (FPR95 < 4.5), with relatively marginal room for improvement compared with other datasets. Yet TIDE still improves upon these already competitive baselines.
>
> > Re Limitation
>
> Due to space constraint we were unable include the limitations in the main text, we will ensure to include them in a future revision.

---

> > ### Author Rebuttal · Reviewer_1QJA · 2026-04-03
> >
> > All my concerns are solved by the authors.

---

> > > ### Author Response · Authors · 2026-04-04
> > >
> > > Dear Reviewer 1QJA,
> > >
> > > We express our greatest gratitude for your insightful feedback and positive recognition. We will ensure to include all clarifications in a future revision.
> > >
> > > We would be very grateful if we can have your kind support during the internal discussions.
> > >
> > > Authors 1142

---

### Official Review · Reviewer_hZWj · 2026-03-13

**Soundness:** 3
**Presentation:** 4
**Significance:** 2
**Originality:** 3
**Overall Recommendation:** 4
**Confidence:** 3

**Summary:**

This paper presents a framework to detect out-of-distribution nodes for node classification tasks in graph learning. The proposed method is based on decomposing input information into feature-specific, structure-specific and joint components and aims to only extract the label-relevant part of the joint information for OOD detection with the belief that incorporating the remaining information leads to overfitting.

**Compliance With Llm Reviewing Policy:**

Affirmed.

**Final Justification:**

My main concern was that scenarios regarding label-relevant feature-specific and structure-specific information were not considered thoroughly in the paper. However, the findings and discussions in the rebuttal answer these questions satisfactorily and elucidate the goal and limitations of TIDE better and thus adds value to the paper. Therefore, I have raised my score.

**Key Questions For Authors:**

1. What are the test sets 'with different difficulty levels' used in Table 1 and how are they constructed? Are these structural, feature or joint shifts in distribution?
2. Is the objective of minimizing pairwise mutual information between the embeddings of  feature-specific, structure-specifc and joint information achieved in practice? For example, how low is the PMI loss component of the overall loss? Does this not in some way counter the IB loss which aims to maximize the joint information?
3. What is meant by label shift in Table 3? More broadly, would it be correct to say that the feature or structural distribution shifts addressed in this paper most likely lead to a change in the decision boundary of the task? What would happen if, there was a 'conceptual' label distribution shift, i.e. nodes with similar features and neighboring structure have different labels, over time lets say. This could lead to for example a different class balance in the network. Is this part of the 'sematic shift' mentioned in section J of the appendix? More importantly, could  a framework such as TIDE detect this?
4. Is it not possible for joint signals to be spurious?

**Limitations:**

Yes. While the limitations are well discussed in the appendix, it may also be worthwhile to briefly mention the critical ones in the main paper as well.

**Strengths And Weaknesses:**

Strengths:

1. A fresh perspective on OOD detection for graphs learning is presented using information decomposition.
2. The paper is well organized and readable despite the framework consisting of multiple overlapping components that are illustrated clearly to ease comprehension.

Weaknesses:

1. If I understand correctly, the core underlying idea upon which the proposed framework is based is that feature-specific and structure-specific information always only have spurious correlations with the label. However, with no evidence provided for this, it seems more like an assumption. For example, what about scenarios where the structural information may be irrelevant, or rather, only the node features are relevant to the label and incorporating the structure is detrimental to the task? [1,2,3] Or perhaps vice versa, when node features are irrelevant and only the structure matters. A justification that these scenarios do not occur is required. One way perhaps could possibly be to show the correlation of the feature and structure specific information with the label in the latent space for several real-world datasets, or the authors could devise a better way. Even if this could be shown empirically, it still seems like a stretch to assume it would always be the case, unless it is theoretically well-motivated. While there is theoretical ground in the paper to show that the proposed objective improves OOD detection, it is unclear how this impacts the in-distribution classification generalization.
2. Several hyperparameters are introduced that require tuning.



[1] When Are Graph Neural Networks Better Than Structure-Agnostic Methods? *ICBINW @ NeurIPS 2022).

[2]  Graph Neural Networks Use Graphs When They Shouldn't (ICML 2024)

[3] GATE: How to Keep Out Intrusive Neighbors (ICML 2024)

---

> ### Author Rebuttal · Authors · 2026-03-26
>
> We appreciate the effort dedicated to providing us with constructive comments. We carefully addressed your concerns as follows.
>
> >W1 Clarification on TIDE design
>
> We clarify that **TIDE does not assume** feature- or structure-specific information is always spurious. Our paper studies the more **fundamental node-level OOD setting**, where shift may arise from features, structure, or both: $(X_{OOD},A_{ID})$, $(X_{ID},A_{OOD})$, or $(X_{OOD},A_{OOD})$. Our motivation is therefore that in prior graph OOD work and our benchmarks, prediction typically depends on both $X$ and/or $A$, while OOD may affect either or both.
>
> **Following the reviewer’s suggestion**, we also tested extreme **single-input-perturbated** settings on Cora and Citeseer (e.g., training on feature-shifted, evaluates on structure/joint-shift data). In such cases, the SL baseline achieved **below 30% ID Acc** and **over 98% FPR95**, suggesting these datasets are not well suited to a purely single-input predictive regime and **do not reflect the main setting of our paper**.
>
> Even in this extreme case, TIDE remains meaningful. If structure provides no predictive information, the joint detector $Z$ will naturally derive limited benefit. This as a failure. Instead, it is an informative outcome as it:
> - provides interpretability into the relative predictive roles of $X$ and $A$ through our tri-network design (Figure 2)
> - can still expose changes in the learned feature-structure relationship under the described OOD cases.
>
> Overall, our focus is the more common graph learning regime, consistent with prior work and the adopted benchmarks. Importantly, TIDE does not harm standard ID prediction: **Table 1** shows competitive ID accuracy with substantially improved OOD detection.
>
> >W2 Hyperparameter tuning
>
> We agree that TIDE introduces additional coefficients beyond a standard SL baseline. However, the setup remains relatively controlled, and we report extensive hyperparameter sensitivity analyses in Appendix L Tables 6 to 8, where TIDE exhibits consistent performance across a suitable range of $\beta, \alpha, \lambda$.
>
> >Q1 On test construction
>
> Yes, this refers to the different OOD scenarios (Sec. 6 and Appendix M) via structure manipulation, feature interpolation, label exclusion, temporal splits (Arvix), and cross-region shifts (Twitch), which **naturally vary in difficulty**. Table 1 reports the average performance on different OOD scenearios for each dataset. We will improve clarity in future revision.
>
> > Q2 PMI clarification
>
> Our intention is not to drive the PMI terms to zero absolutely, but to reduce redundant overlap so that predictive information is allocated to the intended network. This is why Eq. 11) is paired with the network-specific IB objectives in Eq. 12. We have clarified that these terms are complementary than conflicting (Line 247): IB preserves label-relevant content, while PMI discourages the same information from being **redundantly encoded** in multiple networks (where PMI decreases). The ablation in Table 4 supports this: IB alone improves strongly over SL, and adding PMI further improves performance, which will be inconsistent if there is a conflict between PMI and IB.
>
> >Q3 Clarification on Label shift
>
> We follow widely used setting of **label-leave-out** (indeed sematic shift in Appendix J - we will revise this for clarity).
>
> **Re the described scenario**, Sec. 3 defines OOD as either $P_{tr}(X, A) \neq P_{te}(X, A)$ or $P_{tr}(y \mid X, A) \neq P_{te}(y \mid X, A)$, so concept changes in the conditional label semantics are possible conceptually. Our current benchmarks however, mainly evaluate this through label-leave-out/temporal/cross-region protocols, rather than isolated prior-shift.
>
> **Could TIDE detect such conceptual shifts?** In principle, yes, to the extent that the joint representation $Z$ becomes less compatible with the ID decision rule and therefore yields higher uncertainty. This is shown in Prop 5.3: under OOD, the compressed representation contains more irrelevant content relative to ID, producing more separable energy scores.
>
> >Q4 “Is it not possible for joint signals to be spurious?”
>
> In principle, joint signals can be spurious. However TIDE shows that compared with feature- or structure-only cues, **jointly supported signals are better suited for node-level OOD detection**, especially when distribution shift affects both. **TIDE does not assume that all information in the joint branch is trustworthy**: this is why we combine the decomposition with an IB, which aims to retain the label-relevant joint signal while filtering irrelevant content. Thus, our position is not that joint signals can never be spurious, but that **TIDE is designed to learn the shift-indicative label-relevant component**.
>
> Importantly, we also evaluate real-world joint shifts, including temporal shift on Arxiv and the multi-graph Twitch dataset, both are well-motivated cases where feature and structure distributions change together.

---

> > ### Author Rebuttal · Reviewer_hZWj · 2026-04-03
> >
> > Thank you to the authors for their response. However, I am still not convinced that enforcing 'joint information' to be used to learn the decision boundary is a good idea, specially if one of 'single inputs', i.e. node features or structures are irrelevant. Since this is unlikely to be known apriori, I think it can not be considered a 'setting' in which one would know whether TIDE wound be useful or not. It seems that the interpretability mentioned as an advantage in such cases can also come from other models such as MLP vs GNN vs and doesn't require TIDE per se. If I understand correctly, do the authors mean that if the feature or structure specific information is not spurious, it is always going to be contained in the joint information as well? Could the authors elaborate on this and what surety is available for this?

---

> > > ### Author Response · Authors · 2026-04-04
> > >
> > > Dear Reviewer hZWj,
> > >
> > > We thank you for the important follow-up. We address them as follow, in summary:
> > > 1. On "surety": TIDE is **designed to *optimise* toward** joint-input, label-indicative signals, **not to assert** that all non-spurious single-input information must be contained in the joint component. **Under ideal (perfect) optimisation**, all non-spurious label-relevant information from both inputs are indeed expected to be absorbed by the joint-representation $Z$. **In practice** however, at best, the optimisation encourages this as much as possible.
> > > 3. When **one modality is irrelevant**, the tables below show this is a **broader issue of graph-based OOD detection rather than a flaw of TIDE only**.
> > > 4. TIDE is most useful in the graph OOD regime studied in our paper, where both node features and structure contribute to prediction and their mismatch under shift is the central challenge. This is **practical and appears in the studied real graph datasets**.
> > >
> > > ---
> > > ## Toy study: Perfect feature-only scenario
> > >
> > > In an ideal scenario, where **only X is correlated with Y**, while A is noise. e.g.,
> > >
> > > **ID - $\mu$ = 2, std = 1, p = 0.01:** $y_i \sim \text{Bern}(0.5),x_i \mid y_i = c \sim \mathcal{N}(\pm \mu, \sigma^2), A_{ij} \sim \text{Bern}(p)$
> > >
> > > **Feature shift - $\mu_{test}$ = 4, std_test = 1:** $x_i \mid y_i = c \sim \mathcal{N}(\pm \mu_{\text{test}}, \sigma_{\text{test}}^2)$
> > >
> > > **Structure shift - p_test = 0.08:** $A_{ij} \sim \text{Bern}(p_{\text{test}})$
> > > |Perfect Feature Shift|ID ACC|AUROC|AUPR|FPR95|
> > > |---|---|---|---|---|
> > > |MLP|100|100|100|0|
> > > || |*No prop*|||
> > > |SL GNN|62.29|26.31|28.71|99.85|
> > > |TIDE|53.22|**54.70**|**45.32**|**86.63**|
> > > || |*With prop*|||
> > > |SL GNN (GNNSAFE)|62.29|55.63|43.42|75.96|
> > > |TIDE|53.22|**65.80**|**61.41**|**64.92**|
> > >
> > > Prop. =  energy propagation for OOD detection.
> > >
> > > These results show that when the task is truly single-modality, **graph methods are fundamentally mismatched**, because they still learn from both $X$ and the noisy $A$. The **MLP is clearly the optimal model** here. While ID ACC of TIDE slightly falls short (expected in a perfectly single-shift setting), **TIDE improves OOD detection over standard GNNs** by better exploiting the mismatch between learned $X_{ID}-A_{ID}$ interaction from the joint-input optimisation (e.g., **reduces reliance on single-input shortcut to amplify the $X_{OOD}-A_{ID}$ mismatch** ).
> > > |Perfect Struct. shift|ID ACC|AUROC|AUPR|FPR95|
> > > |-|-|-|-|-|
> > > |MLP (Same ID Feature)|100|-|-|-|
> > > |||*No prop*|||
> > > |SL GNN|62.29|36.42|47.25|98.83|
> > > |TIDE|53.22|**64.50**|**62.93**|**69.26**|
> > > |||*With prop*|||
> > > |SL GNN (GNNSAFE)|62.29|77.75|72.21|25.12|
> > > |TIDE|53.22|**87.85**|**85.90**|**15.78**|
> > >
> > > Similarly, under perfect structure shift, GNNs naturally exhibits better performance when energy propagation is applied. Nonethelss, **TIDE performs strongly even without propagation**, again benefiting from joint-input optimisation to exploit the mismatch.
> > >
> > > ---
> > > ## Extreme single-input relevance on real graphs:
> > > We train with noisy structure to mimic a **"feature relevant only"** scenario, and evaluates on structure shifted data. In such cases, both standard GNN/GNNSAFE and TIDE achieved below 30% ID Acc and over 97%+ FPR95, indicating that extreme single input **significantly impacts the overall OOD detection task for typical real-world graphs** (suggesting both modality contributes to predictiveness).
> > >
> > > |Extreme Cora Struct. shift|ID ACC|AUROC|AUPR|FPR95|
> > > |-|-|-|-|-|
> > > |||*No prop*|||
> > > |MLP (Feat only)|43.10|-|-|-|
> > > |SL GNN |28.90|23.72|17.37|100.00|
> > > |TIDE |29.10|27.95|18.36|100.00|
> > > |- Q Struct. net.|10.20|39.98|21.07|95.80|
> > > |- V Feat. net.|30.30|21.54|16.07|100.00|
> > > |||*With prop*|||
> > > |SL GNN (GNNSAFE)|28.90|32.18|20.78|99.86|
> > > |TIDE |29.10|39.82|21.61|97.27|
> > > |- Q Struct. net.|10.20|46.61|23.06|86.63|
> > > |- V Feat. net.|30.30|21.54|16.07|100.00|
> > >
> > > When one modality is effectively irrelevant on real graphs, the **difficulty is not only for TIDE**. The whole node-level graph OOD setup becomes ineffective (extremely poor ID Acc). Hence, justying our focus of when**both modalities are meaningfully related to the label** (real graphs studied by OOD detection community), TIDE’s objective toward jointly supported predictive information is beneficial.
> > >
> > > We also present **the auxiliary network's detection performance**. Corroborating with Fig.2, this provides **interpretable mechanisms directly within TIDE**, where we can identify a **clearer structure shift through the structure network (better than SL GNN** given TIDE's objectives to purify the single-modality information).
> > >
> > > This is all **consistent with the standard OOD detection setting**:
> > > |Standard Cora Struct. shift|ACC|AUROC|AUPR|FPR95|
> > > |-|-|-|-|-|
> > > |||*No prop*|||
> > > |SL GNN|77.80|71.08|45.63|87.48|
> > > |TIDE|78.20|**77.54**|**50.87**|**68.13**|
> > > |||*With prop*|||
> > > |SL GNN (GNNSAFE)|77.80|87.23|77.52|75.04|
> > > |TIDE|78.20|**94.98**|**89.09**|**27.40**|
> > >
> > > We hope this addresses your concerns. We would be grateful if you could consider raising your score.

---

### Official Review · Reviewer_jqkP · 2026-03-18

**Soundness:** 2
**Presentation:** 3
**Significance:** 2
**Originality:** 2
**Overall Recommendation:** 3
**Confidence:** 4

**Summary:**

This paper addresses the challenge of OOD detection for node classification on graphs. The core claim of the work is that SL models tend to capture spurious feature-specific or structure-specific correlations with labels, leading to overconfident predictions on OOD samples. To tackle this, the authors propose TIDE, a Tri-Component Information Decomposition framework that explicitly disentangles the predictive information from graph data into three orthogonal components: feature-specific, structure-specific, and joint-input label-relevant information. TIDE leverages an information bottleneck (IB) objective to preserve only the shift-robust joint information, paired with conditional independence and pairwise mutual information (PMI) regularizers to filter spurious single-input signals. The authors conduct experiments on 7 benchmark datasets to validate TIDE’s superiority over existing graph OOD detection baselines.

**Compliance With Llm Reviewing Policy:**

Affirmed.

**Key Questions For Authors:**

Q1. Have you verified that discarding the auxiliary V and Q networks at inference does not lose valid, shift-robust predictive information, and have you compared your current pipeline with one that incorporates V/Q signals?

**Limitations:**

Yes

**Strengths And Weaknesses:**

**Strengths**

S1. The paper proposes a new structural design targeting graph OOD detection tasks, which splits feature-only and structure-only signals from the mixed graph input. This design is built on the observation that mixed feature-structure representations tend to introduce spurious correlations in graph OOD detection, and forms a complete basic processing pipeline for the split and utilization of different types of graph information.

S2. The authors provide theoretical insights into the advantages of the IB objective over standard SL for graph OOD detection.

S3. The experiments cover a diverse set of seven graph benchmarks, including citation networks, e-commerce co-purchase graphs, academic collaboration networks, social networks, and large-scale temporal academic graphs.

**Weaknesses**

W1. The experimental design, particularly the construction of OOD nodes, appears overly artificial and does not seem to reflect realistic application scenarios. As a result, it is difficult to assess the practical relevance and real-world value of the reported experimental results.

W2. In the method, the auxiliary feature-specific network V and structure-specific network Q serve only as training-time regularizers and are entirely removed during inference, leaving the joint encoder Z as the sole representation for downstream OOD detection. This design appears to neglect the intrinsic trade-off between mitigating spurious correlations and retaining informative predictive cues. Consequently, predictive signals captured by V and Q may be discarded, potentially compromising the model’s discriminative capacity and robustness.

W3. Compared with the extensive literature on disentangled representation learning and information bottleneck-based OOD detection, the core design of this paper offers limited incremental technological novelty.

---

> ### Author Rebuttal · Authors · 2026-03-26
>
> Thank you for the comprehensive review, we clarify your concerns as follows.
> >W1 On experimental design
>
> We clarify that our goal is not to propose a new benchmark, but to evaluate under the **standard graph OOD protocols widely used by prior work** such as EERM(ICLR22), GNNSAFE(ICLR23), NODESAFE(ICML24), and GOLD(ICLR25). We stated that, following prior work, OOD nodes are generated via structure manipulation, feature interpolation, label exclusion (Cora/Citeseer/Pubmed...), and **real-world scenarios** like temporal splits (Arxiv) and cross-region graphs (Twitch).
>
> The paper explicitly discusses (Sec 1) why **controlled settings** such as $(X_{OOD},A_{ID}), (X_{ID},A_{OOD})$, and $(X_{OOD},A_{OOD})$ are needed. Without such controlled settings, it's not possible to validate the proposed tri-component decomposition.
> >W2 On auxiliary networks
>
> We appreciate this insightful observation. This **design is intentional and principled, not a limitation**.
>
> **1. $V$ and $Q$ are regularisers, not inference-time representations.** TIDE decomposes predictive information into:
> * $Z$: joint, shift-indicative signal
> * $V,Q$: feature- and structure-only residual signals
>
> Thus, $V/Q$ are used to isolate input-specific signals, not to support inference.
>
> **2. Why discard $V$ and $Q$ at inference.** Since OOD failure often comes from reliance on input-specific shortcuts, using $V$ and $Q$ at test time would:
> * reintroduce input-specific correlations
> * reduce sensitivity to $X-A$ mismatch
> * weaken OOD detection
>
> So removing them prevents reliance on spurious signals rather than discarding useful information.
>
> **3. Why $Z$ is sufficient.** TIDE preserves useful information in $Z$ through complementary objectives:
> * Eq.10) CInd: pushes joint label-relevant information into $Z$
> * Eq.11) PMI: prevents $V,Q$ from redundantly encoding information already in $Z$
> * Eq.12) IB: retains label-relevant information while filtering noise
>
> Hence, $Z$ captures the joint predictive signal, while $V,Q$ absorb only residual components.
>
> **4. Empirical support.** The results are consistent with this design:
> * **Tab. 1**: competitive ID accuracy with substantially improved OOD detection
> * **Tab. 4**: full TIDE outperforms partial variants
> * **Fig. 2**: clearer ID/OOD separation than baseline
>
> If removing $V/Q$ discarded useful information, we would expect worse ID or OOD performance, but this is not observed.
> >W3 Novelty against prior work
>
> We **respectfully disagree that TIDE is just another application** of IB or disentanglement to OOD detection. TIDE is a **graph-native, modality-aware framework** motivated by a novel information-theoretic view of node-level graph OOD detection (Fig.1).
>
> Our novelty is twofold. First, we introduce a new predictive information decomposition for **graph OOD detection**, instead of  generic disentanglement. Second, we establishe a **theory-to-practice link** for logit/energy-based OOD detection: Sec 5 shows that IB improves ID confidence and enlarges the ID-OOD entropy gap, yielding better energy separation.
>
> This is distinct from prior IB-for-OOD work. [1] studies class-conditional IB in Euclidean settings, without graph structure decomposition or energy-based analysis. [2] also works on Euclidean data and learns different representations under a different objective. [3] studies overconfidence only under the standard SL objective, whereas we directly connect IB to OOD detection.
>
> Finally, TIDE is **not just an architectural tweak**. Table 3 shows that adding the TIDE objective to advanced graph OOD detectors like DeGEM and GOLD often improves detection, suggesting that TIDE offers a transferable training principle instead of a one-off design.
> >Q1 Explanation of removing $V/Q$ at inference
>
> Yes. The evidence indicates that removing $V$ and $Q$ at inference does not harm the predictive signal TIDE aims to preserve.
>
> Prop. 4.1 and Eq. 8) motivate separating the joint label-relevant signal from feature- and structure-specific residual information, and Eq. 9) realises this via a tri-netowork design. Thus, the auxiliary $V/Q$  helps to isolate single-input cues so the shift-indicative signal is concentrated in $Z$.
>
> Fig.2 illustrates the intuition: under structure shift, the $Z$ shows clearer ID-OOD separation, while the feature and structure networks mainly capture residual input-specific patterns. Table 1 shows strong OOD gains with **competitive ID accuracy** using only $Z$, and Table 4 shows the full TIDE objective outperforms partial variants.  Reintroducing $V/Q$ would be inconsistent with their role and adds **unnecessary computation**. We welcome if the reviewer could suggest any potential incorporation.
>
> [1] Li et.al., Task-Oriented Communication with Out-of-Distribution Detection: An Information Bottleneck Framework. GLOBECOM23.
>
> [2] Zhao et.al., Dual Representation Learning for Out-of-distribution Detection. TMLR23.
>
> [3] Hu et.al., An Information Theoretical View for Out-Of-Distribution Detection. ECCV24.

---

> > ### Author Rebuttal · Reviewer_jqkP · 2026-04-03
> >
> > Thanks for your response. However, since you do not provide convincing evidence, I remain concerned that discarding the auxiliary V and Q networks at inference time may remove useful and shift-robust predictive information. This could have a substantial negative impact on the model’s predictive performance.

---

> > > ### Author Response · Authors · 2026-04-04
> > >
> > > Dear Reviewer jqkP,
> > >
> > > Thank you for the followup. To **directly answer your concern on whether discarding V and Q removes useful and shift-robust predictive information**, we conducted an inference-time ablation over **Z, Q, V, Z+Q, Z+V, and Z+V+Q**, where network-specific detection scores are computed independently and combined via an unweighted sum of logits. This serves as a **controlled diagnostic probe** rather than a new inference strategy.
> > >
> > > ---
> > >
> > > **Summary:** Discarding the auxiliary networks V and Q at inference time **does not** remove useful, shift-robust predictive information and does not hurt predictive performance.
> > >
> > > ---
> > >
> > > Across the four reported settings below (Cora structure shift, Citeseer feature shift, Twitch cross-graph shift, and Coauthor label shift), the results show a **consistent and robust pattern**:
> > >
> > > 1. **Z alone achieves the best OOD performance and competitive ID accuracy overall.**
> > > For example, on Cora, Z reaches FPR95 27.40 vs. Z+Q: 29.85, Z+V: 38.04 and Z+V+Q:37.48. On Twitch, the gap is even larger (46.60 vs. 82.42 and 84.54 and 93.50). This indicates that adding the auxiliary networks back again would degrade performance, validating our TIDE design for efficient information optimisation.
> > >
> > > 2. **Adding Q and/or V consistently degrades OOD detection.**
> > > Across all datasets, Z+Q, Z+V, and Z+V+Q lead to lower AUROC/AUPR and substantially worse FPR95. This degradation is particularly pronounced under stronger shifts (e.g., Twitch), where combining networks can collapse performance (e.g., AUROC drops from 89.72 to 52.86 for Z+V+Q).
> > >
> > > 3. **Q and V alone are not reliable predictors.**
> > > Both networks exhibit very poor standalone performance: for instance, on Cora, Q and V drop to 27.30 and 38.90 Acc, and on Citeseer the AUROC they are close to random  (50.88 and 59.29). This indicates that they do not encode sufficient label-relevant or shift-robust information on their own, which is **expected** as our tri-network optimisation aims to prioritise useful information to be learned by Z.
> > >
> > > 4. **No consistent ID accuracy benefit from using auxiliary networks.**
> > > The ID accuracy of V/Q alone are limited (consistent with the effect of our regularisers), they do not improve the ID accuracy and OOD detection in a consistent way when incorporating with Z. Notably, the Z network typically performs better alone (suggesting sufficient and relevant label-predictive information being captured), further indicating limited useful predictive contribution of using V and Q.
> > >
> > > Taken together, these results provide strong and consistent empirical evidence that **V and Q do not contain additional shift-robust predictive information beyond Z**. If such information were present, incorporating these branches at inference should improve performance under distribution shift. However, **we observe the opposite effect across all evaluated settings**.
> > >
> > > **These findings support our design**: V and Q primarily capture input-specific residual signals, which are useful during training for disentanglement and regularisation, but are not beneficial for inference. Retaining them at test time introduces spurious or shift-sensitive information, thereby degrading robustness and wasting unnecessary computation (~2-3x inference latency).
> > >
> > > We hope this addresses your concerns. We would be grateful if you could consider raising your rating.
> > >
> > > |Cora Struct. shift|ID ACC|AUROC|AUPR|FPR95|
> > > |-|-|--|-|-|
> > > |GNNSAFE|77.80|87.23|77.52|75.04|
> > > |TIDE ($Z$ Classifier network)|78.20|**94.98**|**89.09**|**27.40**|
> > > |-$Q$ Structure network|27.30|55.77|47.78|88.84|
> > > |-$V$ Feature network|38.90|40.24|21.98|95.49|
> > > |-$Z+Q$|78.40|91.88|86.90|29.85|
> > > |-$Z+V$|75.20|89.38|69.95|38.04|
> > > |-$Z+V+Q$|77.60|89.88|70.24|37.48|
> > >
> > > |Citeseer Feat. shift|ID ACC|AUROC|AUPR|FPR95|
> > > |-|-|-|-|-|
> > > |GNNSAFE|64.90|84.48|68.57|66.31|
> > > |TIDE ($Z$ Classifier network)|70.10|**93.30**|**83.66**|**27.23**|
> > > |-$Q$ Structure network|17.30|50.88|23.06|95.46|
> > > |-$V$ Feature network|32.60|59.29|27.38|90.02|
> > > |-$Z+Q$|68.80|90.61|70.06|32.85|
> > > |-$Z+V$|68.60|90.18|80.32|30.67|
> > > |-$Z+V+Q$|67.60|90.51|70.59|34.90|
> > >
> > > |Twitch (3 Test graphs)|ID ACC|AUROC|AUPR|FPR95|
> > > |-|-|-|-|-|
> > > |GNNSAFE|70.75|66.33|72.59|81.18|
> > > |TIDE ($Z$ Classifier network)|68.63|**89.72**|**91.92**|**46.60**|
> > > |-$Q$ Structure network|57.70|52.09|63.37|95.45|
> > > |-$V$ Feature network|62.87|50.95|60.44|94.93|
> > > |-$Z+Q$|64.73|85.11|91.45|82.42|
> > > |-$Z+V$|63.52|73.47|80.09|84.54|
> > > |-$Z+V+Q$|56.99|52.86|64.02|93.50|
> > >
> > > | Coauthor Label Shift | ID ACC| AUROC | AUPR | FPR95 |
> > > | - | - | - | - | - |
> > > | GNNSAFE | 95.23 | 97.62| 99.13|11.07  |
> > > | TIDE ($Z$ Classifier network) | 95.45 | **97.94**   | **99.25** | **8.93** |
> > > | - $Q$ Structure network   |30.14 | 50.20 | 73.93   | 94.33 |
> > > | - $V$ Feature network  |  75.74 | 84.36| 90.07 | 53.27 |
> > > | - $Z+Q$   | 93.03 | 95.35 | 98.29 | 25.10 |
> > > | - $Z+V$  | 94.93 | 96.31   |98.14 | 11.56 |
> > > | - $Z+V+Q$  | 93.54 | 96.00 | 98.60 | 22.69 |

---

### Decision · Program_Chairs · 2026-04-30

**Decision:**

Accept (regular)

**Comment:**

The paper studies graph OOD detection through a tri-component information decomposition framework. Reviewers agreed that the paper is clear, technically interesting, and supported by the experiments. Several reviewers found the information-theoretic perspective useful, especially the analysis connecting IB training and OOD separation. The main concerns were about the strength of the underlying assumptions, the realism of some OOD settings, and the theoretical justification for the surrogate decomposition. I have read the rebuttal and incorporated the response into my decision. The rebuttal addressed most technical concerns. Finally, the discussion reflects divergent opinions.